# L-DOS47 Elevates Pancreatic Cancer Tumor pH and Enhances Response to Immunotherapy

**DOI:** 10.3390/biomedicines12020461

**Published:** 2024-02-19

**Authors:** Bruna Victorasso Jardim-Perassi, Pietro Irrera, Oluwaseyi E. Oluwatola, Dominique Abrahams, Veronica C. Estrella, Bryce Ordway, Samantha R. Byrne, Andrew A. Ojeda, Christopher J. Whelan, Jongphil Kim, Matthew S. Beatty, Sultan Damgaci-Erturk, Dario Livio Longo, Kim J. Gaspar, Gabrielle M. Siegers, Barbara A. Centeno, Justin Y. C. Lau, Shari A. Pilon-Thomas, Arig Ibrahim-Hashim, Robert J. Gillies

**Affiliations:** 1Department of Metabolism and Physiology, H. Lee Moffitt Cancer Center and Research Institute, Tampa, FL 33612, USApietro.irrera@moffitt.org (P.I.); bordway@mgh.harvard.edu (B.O.); sultandamgaci@gmail.com (S.D.-E.);; 2Department of Immunology, H. Lee Moffitt Cancer Center and Research Institute, Tampa, FL 33612, USA; oluwaseyi.oluwatola@moffitt.org (O.E.O.);; 3Department of Molecular Medicine, University of South Florida, Tampa, FL 33620, USA; 4Comparative Medicine, University of South Florida, Tampa, FL 33620, USA; 5Massachusetts General Hospital, Harvard Medical School, Boston, MA 02115, USA; 6Department of Biological Sciences, University of Illinois, Chicago, IL 60607, USA; 7Department of Biostatistics and Bioinformatics, H. Lee Moffitt Cancer Center and Research Institute, Tampa, FL 33612, USA; 8Institute of Biostructures and Bioimaging (IBB), National Research Council of Italy (CNR), 10126 Turin, Italy; 9Helix BioPharma Corp., Bay Adelaide Centre-North Tower, 40 Temperance Street, Suite 2700, Toronto, ON M5H 0B4, Canada; 10Department of Pathology, H. Lee Moffitt Cancer Center and Research Institute, Tampa, FL 33612, USA; 11Small Animal Imaging Laboratory (SAIL), H. Lee Moffitt Cancer Center and Research Institute, Tampa, FL 33612, USA; drjustinlau@gmail.com

**Keywords:** pancreatic cancer, targeted therapy, L-DOS47, acidosis, immunotherapy

## Abstract

Acidosis is an important immunosuppressive mechanism that leads to tumor growth. Therefore, we investigated the neutralization of tumor acidity to improve immunotherapy response. L-DOS47, a new targeted urease immunoconjugate designed to neutralize tumor acidity, has been well tolerated in phase I/IIa trials. L-DOS47 binds to CEACAM6, a cell-surface protein that is highly expressed in gastrointestinal cancers, allowing urease to cleave endogenous urea into two NH4+ and one CO_2_, thereby raising local pH. To test the synergetic effect of neutralizing tumor acidity with immunotherapy, we developed a pancreatic orthotopic murine tumor model (KPC961) expressing human CEACAM6. Using chemical exchange saturation transfer–magnetic resonance imaging (CEST-MRI) to measure the tumor extracellular pH (pHe), we confirmed that L-DOS47 raises the tumor pHe from 4 h to 96 h post injection in acidic tumors (average increase of 0.13 units). Additional studies showed that combining L-DOS47 with anti-PD1 significantly increases the efficacy of the anti-PD1 monotherapy, reducing tumor growth for up to 4 weeks.

## 1. Introduction

Acidosis is a well-established feature of cancer, favoring its progression and metastatic spread. Tumor acidosis is an extracellular condition provoked by an increased production of acidic molecules and protons, and it alters neoplastic tissues’ physiological homeostasis, impacting many cell subtypes, including macrophages, T and B cells, epithelial cells, and fibroblasts [1,2]. In healthy conditions, acidosis is also present, particularly in lymph nodes where T cells sustain an acidic environment to suppress their own functions and become active only after they have left the acidic lymph node site [3]. As such, immune suppression and deregulation represent some of the crucial processes to be addressed when treating cancer because these events occur as physiological responses to an acidosis condition [4].

An acidic environment can contribute to immune impairment, wherein many components, such as macrophages [5], T cells [6], and natural killer (NK) cells, shift to a state that supports tumor growth [7]. Therapies inhibiting these pathways can restore immune responses against cancer; however, if the underlying acidic condition remains, these effects are lost even when combined with immunotherapies [5]. 

Immune treatments and inhibitory strategies alone are often insufficient to overcome the effects of acidosis. Directly counteracting acidity through alkalinization treatments is a more reliable approach, and coupling buffer therapies with immune/inhibitory treatments may result in better outcomes [8,9,10]. Studies have shown how this strategy could be applied in several tumor models: combining acidity-lowering drugs with chemotherapy was essential to overcome chemoresistance in human and murine tumors [11,12,13,14], and chronic oral administration of bicarbonate coupled with immune therapy significantly reduced tumor growth compared to monotherapies alone [15]. Yet, even though these approaches have been proven successful in the preclinical setting, a direct translation into the clinic setting is often delayed or unfeasible because they are either not effective in patients [16] or not safe enough for long-term use. 

In the present study, we used a combination approach involving L-DOS47, a new pH-targeting molecule developed by Helix BioPharma (Toronto, ON, Canada), in addition to the canonical immune therapy provided by the administration of an anti-programmed death (PD1) antibody. L-DOS47 is an immunoconjugate comprising multiple copies of a camelid single-domain antibody that specifically binds the carcinoembryonic antigen-related cell adhesion molecule 6 (CEACAM6), which is constitutively upregulated in human cancer cells [17,18], and is conjugated to a jack bean-derived urease enzyme [19]. Its mechanism of action involves selective binding to CEACAM6 on the tumor cell surface, thereby localizing urease, which converts endogenous urea into NH_3_ and CO_2_ with a net production of bicarbonate and hydroxyl ions and causes alkalinization of the extracellular tumor microenvironment (TME). In phase I clinical trials, L-DOS47 was well tolerated in patients with non-small-cell lung cancer (NSCLC) and yielded encouraging results when combined with pemetrexed plus carboplatin, with 75% of patients showing clinical benefits (stable disease, complete response, or partial response) [20]. Since L-DOS47 is suitable for cancers that express high levels of its target antigen CEACAM6, pancreatic and gastrointestinal cancers are other potential applications in addition to lung cancer [21,22].

Pancreatic ductal adenocarcinoma (PDAC) remains a deadly cancer, with dismal survival rates necessitating the development of more effective therapies. PDAC has been historically considered immunologically “cold”, yet a growing number of studies have indicated the inherent heterogeneity and potential reversibility of this phenotype [23]. Attempting to improve outcomes in PDAC by utilizing a combination of checkpoint inhibitors with therapies targeting immunosuppressive features, such as acidosis in the tumor microenvironment, is a logical next step. To this end, we generated a preclinical pancreatic tumor model to investigate the ability of L-DOS47 to increase tumor pH and thereby control tumor growth. To measure tumor pH in vivo, we used the emerging non-invasive chemical exchange saturation transfer (CEST) technique coupled with iopamidol injection, which can detect water exchange rate changes that correlate with pH changes. Iopamidol was first developed and used for CT imaging but was later applied as an MR contrast agent since the amide groups in its structure can interact (exchange) with water in a pH-dependent manner [24]. Many previous preclinical studies have used this approach to correlate the acidic state of a tumor with its aggressiveness and invasiveness or to evaluate treatment response and the onset of resistance [25,26,27,28], thereby establishing the foundations for clinical applications [29,30,31].

Using CEST-MRI tumor pH mapping in our PDAC model, we confirmed that L-DOS47 raises the pHe of acidic tumors, which contributes to the enhanced efficacy of anti-PD1 therapy in combination with L-DOS47.

## 2. Materials and Methods

### 2.1. Transduction and Selection of CEACAM6-Expressing Clone

The murine pancreatic cancer cell line UN-KPC-961 (KPC961) was obtained via MTA from Dr. Surinder K. Batra (University of Nebraska Medical Center, Omaha, NE, USA) [32]. This cell line was chosen because it has a similar expression and mutation pattern (p53^R172H^/KRas^G12D^) to human PDAC cells that were retrovirally infected with hCEACAM6 pLenti-GIII-EF1a lentivirus, and a CEACAM6-expressing subclone was selected for this study. For transduction, cells were trypsinized, centrifuged, and diluted to a concentration of 50 K cells per mL in a DMEM/F12 medium (Gibco, Waltham, MA, USA) supplemented with 10% FBS, 1% of P/S (Sigma, St. Louis, MO, USA), and 5 µg/mL polybrene. A total of 8 µL of the CEACAM6 lentivirus was added to 500 µL of complete DMEM/F12, and the lentiviral mixture was added to the wells of a 6-well plate. Then, 1 mL of the previously diluted cells was added on top of the lentiviral mixture in the 6-well plate. At 18 h after the addition of the cells, the medium was removed, and cells were cultured in complete DMEM/F12. After 24 h, the medium was replaced with complete DMEM/F12 containing 5 µg/mL puromycin (Gibco, Waltham, MA, USA). Once at 90% confluence in the 6-well plate, cells were trypsinized and transferred to a T-75 flask to be maintained in an incubator at 37 °C and 5% CO_2_. 

For the selection process, cells were trypsinized, centrifuged, diluted, counted, and then passed through a 4 µm filter. Cells were diluted and counted again, and then diluted once more to achieve a final concentration of 500 cells per mL. A total of 2 μL of the diluted cells was added to each well of 24-well plates. Drops were assessed for the presence of cells, with wells containing more than one cell being omitted. All wells were then filled with DMEM/F12 containing 5 µg/mL puromycin. After 24 h and again after 120 h, wells were visually assessed to determine which wells contained single colonies of cells. Wells containing no cells or multiple colonies were omitted. Each clone (single-cell origin) was transferred to 6-well plates to be expanded. Once confluent, the cells were collected and the expression of human CEACAM6 was verified via flow cytometry. 

### 2.2. Flow Cytometry

#### 2.2.1. CEACAM6 Expression

KPC961-CEACAM6-transduced subclones and KPC961 parental cells (negative control) were resuspended in FACS buffer (PBS with 5% FBS, 1 mM EDTA, and 0.1% Sodium Azide) to a concentration of 0.5–1 × 10^6^ cells/mL for flow cytometric analysis. Cells were stained in FACS buffer with anti-CEACAM6-FITC antibody (Sino Biological 10823-R408R, Sino Biological, Inc., Wayne, PA, USA) at 10 µg/mL for 30 min at 4 °C in the dark. After incubation, cells were washed with FACS buffer, centrifuged, and resuspended in FACS buffer containing 1.25 µg/mL live/dead PI reagent (Bioscience Propidium Iodide—Fisher 5018262, Fisher Scientific, Waltham, MA, USA). Flow cytometry data were acquired using the BD FACSCelesta (BD Biosciences, Franklin Lakes, NJ) and analyzed using FlowJo ver10.8.1 (Tree Star, Ashland, OR, USA). Gating strategy is presented in Appendix A.

#### 2.2.2. Tumor Digestion and Single-Cell Preparation for Flow Cytometry Analysis

Single-cell suspensions were prepared from KPC961-1B6 tumors by first cutting the tumors into minute fragments. These fragments were placed in a GentleMACS C-tube and processed by enzymatic digestion in HBSS (Life Technologies, Carlsbad, CA, USA) containing 1 mg/mL Collagenase D, 1 mg/mL DNAse I, and 2.5 mg/mL Hyaluronidase (all from Sigma-Aldrich, Waltham, MA, USA) and dissociated using a GentleMACS dissociator (Miltenyi Biotec, Bergish Gladback, Germany). Following dissociation, the C-tube was stirred in a water bath at 37 °C for 45 min. After stirring, the tissues were passed through the GentleMACS dissociator once more. The resulting suspension was put through a 70 µm cell strainer. Cells were then pelleted via centrifugation; the supernatant was discarded, and a red blood cell lysis buffer (BioLegend, San Diego, CA, USA) was added to remove any RBCs. After RBC lysis, the cells were passed through a 70 µm cell strainer once more. The cells were then washed with PBS, pelleted, and resuspended in FACS buffer (PBS with 5% FBS, 1 mM EDTA, and 0.1% Sodium Azide) at a concentration of 0.5–1 × 106 cells/mL for flow cytometry analysis. The cells were labeled with the following antibodies: CEACAM6-FITC at 10 µg/mL, Sino Biological 10823-R408R; H2Kb-Pac Blue at 0.5 mg/mL, BioLegend 116514; CD45-BV605 at 0.2 mg/mL, BioLegend 103155 (San Diego, CA, USA); and PD-L1—PE at 0.2 mg/mL, Invitrogen 12-5982-82, Waltham, MA, USA) in FACS buffer for 20 min at 4 °C in the dark. Prior to antibody staining, a Fc receptor blocker (Tonbo Biosciences 70-0161-M001, Tonbo Biosciences, San Diego, CA, USA) was added for 10 min at 4 ℃ to prevent non-specific binding of antibodies. Live/dead fixable near-IR reactive dyes (Thermo Fisher Scientific, Waltham, MA, USA) were used to exclude dead cells before analysis. Cell data were acquired using the BD FACSCelesta (BD Biosciences) and analyzed using FlowJo v10.10 (Tree Star). Gating strategy is presented in Appendix A. 

### 2.3. In Vitro Metabolic Profiling

#### 2.3.1. Oxygen Consumption and Extracellular Acidification Measurements

A Seahorse Extracellular Flux (XF-96) analyzer (Seahorse Bioscience, Chicopee, MA, USA) was used to measure real-time basal oxygen consumption (OCR) and extracellular acidification rates (ECAR) in KPC961-1B6 and KPC961 parental cells. The cells were seeded in an XFe-96 microplate (Seahorse, V3-PET, 101,104–004) in a normal growth medium overnight. The growth medium was replaced with a DMEM powder base medium (Sigma D5030) supplemented with 1.85 g/L sodium chloride and 1 mM glutamine, and pH was set to 7.4. When testing glycolysis, cells were incubated in a glucose-free medium and incubated for 1 h in a non-CO_2_ incubator prior to measurement. ECAR and OCR were measured in the absence of glucose associated with the non-glycolytic activity, followed by two sequential injections of D-glucose (6 mM) and oligomycin (1 µM) in real time, which were associated with glycolytic activity (glucose-induced ECAR) and glycolytic capacity (reserve). The mitochondrial stress test was also used whereby cells were incubated in a glucose (5.5 mM) and glutamine (1 mM)-containing medium, and basal OCR and ECAR were measured prior to the sequential injection of oligomycin (1 µM), which was associated with ATP-linked OCR, and FCCP (1 µM) was associated with the mitochondrial reserve capacity and Rotenone/Antimycin A (1 µM). 

Once the Seahorse assay was finalized, cells were stained using a 1:1000 dilution of the HCS NuclearMask Red Stain (Molecular Probes, cat# H10326, Molecular Probes, Eugene, OR, USA). Cells were incubated at 37 °C for 30 min, washed, and later imaged using an Incucyte S3 Live-Cell Analysis System (Sartorius, Gottingen, Germany). Plates were scanned using the 96-well TPP plate setting, magnification was set at 4X, and both the phase and Red FL filters were applied. The Incucyte Basic Analyzer module (version 2022B) with Top-Hat background subtraction and intensity/size thresholding was used to identify Red FL nuclei and determine both cell count and confluency. The OCR and ECAR values were normalized to the cell number using a cell quantification software from Incucyte S3 mentioned above.

#### 2.3.2. Lactate Measurement

Twenty thousand cells were seeded in a 96-well plate in a 200 μL growth medium containing 10% FBS. The medium was collected following a 48 h incubation period and measured for lactate production using a biochemistry analyzer, YSI 2900 (Xylem, Washington, DC, USA). The cell densities per well were determined by means of Incucyte cell count application with the use of a nuclear staining technique. Cells were incubated for 30 min with a 1:1000 dilution of the HCS Nuclear Mask Red stain (Molecular Probes cat# H10326). After washing the cells in 1X PBS, these cells were transferred to the Incucyte S3 Live-Cell Analysis System (Sartorius, Gottingen, Germany) where cell count and confluency were determined. Data were normalized to the cell number and were reported as lactate production in g/L/cells.

### 2.4. KPC961 Orthotopic Tumor Model

The animal experiments were approved by the Institutional Animal Care and Use Committee (IACUC, protocols #8596 and #10942). Mice were obtained from The Jackson Laboratory (Bar Harbor, ME, USA) and housed in a facility under pathogen-free conditions in accordance with the IACUC standards of care at the H. Lee Moffitt Cancer Center. 

A total number of 5 × 10^4^ KPC961-1B6 cells were inoculated orthotopically into the pancreas of B6.129 mice. These mice were dosed with Meloxicam (5 mg/kg) 30 min before surgery to provide analgesia, and isoflurane (2% given in 1.5 L/min oxygen breathing) was utilized to induce anesthesia during the procedure. After removing hair and sterilizing the mice’s midsection, abdominal skin and muscle were incised to allow direct injection of 20 µL bolus of cells/PBS into the head of the pancreas. Closure of the abdominal cavity was accomplished in two layers, and the skin layer was closed in a simple interrupted pattern with non-absorbable surgical staples, which were removed 10 days post-inoculation. The surgical procedure is described in detail in [26].

In the in vivo experiment of our study, we observed a continuous response (tumor growth) in the study subjects that were randomly assigned to multiple treatments following a factorial analysis of variance (ANOVA) experimental design. To estimate sufficient sample size numbers needed per treatment, we assumed the responses we observed in the experimental treatments met the normality assumption with a standard deviation of 10%. Given this assumption and basing the sample size estimates on a matched-pair design, wherein the true effect size (i.e., the difference between control and case) was 10%, we would need at least 10 subject pairs (case and control) to be able to reject the null hypothesis that this response difference was zero with a probability or power (beta) of 80%. The Type 1 error (false positive) probability associated with this test of the null hypothesis (alpha) was 5%. In our study, the number of study subjects in each treatment group in our vivo experiments met or exceeded the minimum sample size, with 39 mice in replicate 1 [control (n = 10), anti-PD-1 (n = 10), L-DOS47 (n = 10), and anti-PD1 + L-DOS47 (n = 9)]; 39 mice in replicate 2 [control (n = 9), anti-PD1 (n = 10), L-DOS47 (n = 8), and anti-PD1 + L-DOS47 (n = 12)]; and 41 mice in replicate 3 [control (n = 7), anti-PD1 (n = 15), L-DOS47 (n = 5), and anti-PD1 + L-DOS47 (n = 14)]. 

### 2.5. CEST Imaging

CEST-MRI was performed to measure extracellular pH (pHe) in the KPC961-1B6 orthotopic tumors at baseline (pre-L-DOS47) and at 4 h, 18 h, 24 h, 48 h, 72 h, and 96 h post L-DOS47L-DOS47 administration. Each mouse was imaged to acquire a baseline tumor pHe (pre-L-DOS47) and at one or two time points after L-DOS47 administration. 

MR images were acquired using a 7T horizontal-bore magnet (Agilent ASR 310, Santa Clara, CA, USA; Bruker Biospin, Inc. BioSpec AV3HD, Billerica, MA, USA), with a 1H 30 mm volume coil (m2m Imaging Corp, Cleveland, OH, USA). CEST was performed with an intraperitoneal (IP) delivery of the contrast agent (iopamidol) as described in [26]. In a previous study, we determined that an interval of 2 days was necessary between CEST-MRI sessions in the same mouse to allow for contrast-agent clearance [26]. Because of this and MRI availability at our facility, the time between pre- and post-L-DOS47 MRI sessions ranged from 2 to 13 days. The mean tumor volumes across all groups were maintained as similar as possible for the baseline (250 ± 97.83 mm^3^) and post-L-DOS47 time points (422.4 ± 155.3 mm^3^).

The mice received an intramuscular injection of xylazine (6 mg/kg, AnaSed, Akorn Pharmaceutical, Chicago, IL, USA) as a muscle relaxant before their placement in the scanner and were kept anesthetized with isoflurane at 2% during MRI scanning. A continuous-wave (CW) irradiation scheme was adopted for the radiofrequency (RF) pre-saturation pulse (3 μT for 5 s), followed by a RARE single shot sequence (TR = 6.0 s, slice thickness = 1 mm, matrix size = 64 × 64, and spatial resolution = 543 µm). This scheme was repeated for every frequency of interest that was contained in the range of −10 to 10 ppm, totaling a final acquisition time of 4 min and 36 s for each CEST scan. The collected images were then analyzed using MATLAB 2021a (The MathWorks, Inc. Natick, MA, USA) with a homemade script wherein a multipool Lorentzian fit extrapolated the signal changes between the pre-contrast and post-contrast images and then subtracted the background signal (pre-contrast) from each of the post-contrast acquisitions. As a result, only the direct contribution of iopamidol was taken into consideration for the final pH measurements, where the ratio between the contributing pools was interpolated with a calibration curve to obtain a pixel-by-pixel pH map. pH values were slightly more prone to error at both ends of the calibration curve, but not in the physiological range that was found in this study. Of the eight post-contrast images, only the last three (30, 35, and 40 min) were used to obtain the averaged pHe for each tumor.

### 2.6. Treatments for Efficacy Study

Three biological replicates were performed for efficacy studies. For each experiment, B6.129 mice were inoculated orthotopically into the pancreas with KPC961-1B6 cells. Six days after tumor inoculation, tumor volumes were measured via ultrasound imaging, and the mice were randomized into four groups with equal tumor volume averages to initiate therapies. Mice that developed an additional tumor in the abdominal wall were excluded from the study. The treatment groups included (1) control (no therapy); (2) monotherapy with anti-PD1; (3) monotherapy with L-DOS47; and (4) anti-PD1 + L-DOS47 combination therapy. Treatments were administrated twice a week. Anti-PD1 (InVivoMab anti-mouse PD-1 (CD279), Clone: RMP1-14, Isotype: rat IgG2a, Bio X Cell, Lebanon, NH) was administered IP at a dose of 300 µg, while L-DOS47 was administered intravenously at a dose of 90 μg/kg. The mice in the anti-PD1 + L-DOS47 combination group were treated with anti-PD1 (300 µg) first and then L-DOS47 (90 μg/kg) 4 h later. 

The mice used in the biological replicates were as follows: 39 mice in replicate 1 [control (n = 10), anti-PD-1 (n = 10), L-DOS47 (n = 10), and anti-PD1 + L-DOS47 (n = 9)]; 39 mice in replicate 2 [control (n = 9), anti-PD1 (n = 10), L-DOS47 (n = 8), and anti-PD1 + L-DOS47 (n = 12)]; and 41 mice in replicate 3 [control (n = 7), anti-PD1 (n = 15), L-DOS47 (n = 5), and anti-PD1 + L-DOS47 (n = 14)]. 

Tumor volumes were measured weekly via ultrasound imaging, and tumor growth was monitored for 4 weeks. Mice with tumors that reached or exceeded 750 mm^3^ (end-point tumor volume) were humanely euthanized. 

### 2.7. Ultrasound Imaging

The mice were imaged using the Vevo 2100 ultrasound system (FUJIFILM VisualSonics Inc., Toronto, ON, Canada) to measure the volumes of the orthotopic pancreatic tumors. The mice were anesthetized with 1.5–3% isoflurane delivered via nose-cone manifold, depilated with Nair, and positioned with a surgical tape onto a thermo-regulated stage where the electrodes and rectal probe continuously monitored their body temperature, heart rate, and respiration rate. An adjustable heat lamp and a pre-warmed ultrasound gel were used to ensure that the animals maintained their body temperature during scanning. Scans were conducted at thicknesses of 0.05 mm with the 3D motor attachment. The regions of interest (ROIs) were obtained from parallel slices to measure the tumor volume using the Vevo LAB 5.5.0 software.

### 2.8. Statistical Analyses

To determine how pHe varied across the tumors in response to L-DOS47 administration, we constructed Empirical Cumulative Distribution Functions (ECDFs) of pHe pixel distributions in the tumor ROIs imaged using CEST-MRI before and after L-DOS47 administration. The ECDF curves were obtained from the combined pHe measurements using package ggplot2 in the R programming language version 4.0.0 22. We then tested whether the ECDF curves differed significantly by means of a Kolmogorov–Smirnov test using the R package dgof. We used the “jitter” command to eliminate tied pHe values. Statistical significance was assessed at *p* ≤ 0.05.

Linear mixed-effects models were used to test for differences in tumor growth rates among the treatment arms up to 4 weeks. The tumor volume measurements were obtained on days 6–7 (week 1), 13–14 (week 2), 20–21 (week 3), and 27–28 (week 4). A tumor volume of 750 mm^3^ was considered the “endpoint”; thus, if a mouse reached the endpoint tumor volume before the end of the experiment, from that time until the experiment was concluded, the tumor volume used for analyses was recorded at 750 mm3. Statistical analysis was carried out using package LME4 and package lmerTEST in the R programming language. Post hoc pairwise tests of the estimated marginal means based on the linear mixed-effects model were conducted using the R emmeans package. Data were transformed with log base 2 to linearize the growth rate trends. Data from replicates 1, 2, and 3 were combined, and we modeled log2 (tumor growth) as the dependent variable, day and treatment arm as the fixed main effects, the interaction of day and treatment arm, and mouse as the random effect (random intercept). 

## 3. Results

### 3.1. Development of an Orthotopic Pancreatic Tumor Model That Expresses Human CEACAM6

An orthotopic pancreatic tumor model was generated using the murine PDAC cell line KPC961, which was engineered to express human CEACAM6 (hCEACAM6). After transduction, flow cytometric analyses confirmed the presence of 97.7% hCEACAM6-expressing cells in clone 1B6 (Figure 1a and Appendix A). Metabolic profiling showed no differences in energy metabolism between the KPC961 parental and clone 1B6 cells (Figure 1b,c).

We additionally confirmed that the KPC961 clone 1B6 could form orthotopic tumors in immunocompetent B6.129 mice (Figure 1d) and showed continued CEACAM6 expression (27.2 ± 3.56%) in inoculated tumors as indicated by flow cytometry (Figure 1e and Appendix A). These tumors were also tested for the presence of the major histocompatibility complex (MHC) class I molecule H2K^b^ and PD-1 ligand 1 (PD-L1), which represented 75.2 ± 9.8% and 21.33 ± 4.45% of the tumor cells, respectively (Figure 1e and Appendix A). 

### 3.2. L-DOS47 Increases Tumor Extracellular pH in Acidic Tumors

We next performed pharmacodynamic studies using CEST-MRI to measure the time-dependent changes in pHe following L-DOS47 administration in mice bearing KPC961 clone 1B6 orthotopic tumors. In this approach, Z-spectra were calculated inside the region of interest (ROI) that was drawn on an anatomical image reference corresponding to the tumor region, followed by a fitting process that calculated the ratiometric values to obtain the pHe tumor map (Figure 2a). 

First, a baseline tumor pHe was obtained before the administration of L-DOS47 for all mice. Then, each mouse received a single bolus injection of L-DOS47 (90 µg/kg) (Figure 2b). Tumor pHe was measured via CEST MRI at one or two time points following the L-DOS47 administration (4 h, 18 h, 24 h, 48 h, 72 h, and 96 h). 

Figure 2c–f show representative pixel-wise pHe maps and histograms of tumor pHe distribution for individual mice at each pre- and post-L-DOS47 time point. Following the administration of L-DOS47, the pHe tumor maps showed pixels that represented higher pHe values (Figure 2c,d), and the histograms confirmed that the pHe values shifted to the right of the pH axis in the histograms (Figure 2e,f), indicating an increase in pHe across the tumor ROI for all time points post L-DOS47 administration. 

However, this increase in tumor pHe was evident only in tumors with acidic pHe at baseline (pre-L-DOS47) (Figure 3). Since the baseline tumor pHe differed among the animals, we analyzed changes in the mean pHe before and after L-DOS47 treatment for each mouse individually, rather than averaging all mice for each time point. The changes in tumor pHe before and after L-DOS47 administration (delta of the mean tumor pHe) indicated that a single dose of L-DOS47 could increase pHe in tumors with a baseline pHe ≤ 6.60, but not in those with a baseline pH > 6.60 (Figure 3a,b).

Further statistical analyses of the pHe histogram distributions confirmed that acidic tumors showed significantly higher pHe values at all time points post L-DOS47 administration (Figure 3c), except at 18 h post L-DOS47 administration, which showed a similar trend but did not reach significance. A single dose of L-DOS47 did not increase the pHe in tumors with a baseline pHe > 6.6 (Figure 3c, Appendix A).

### 3.3. L-DOS47 Has a Synergistic Effect on Anti-PD1 Therapy in Reducing Tumor Growth

Once the pharmacodynamic studies had confirmed that L-DOS47 increases tumor pHe in the KPC961-1B6 orthotopic model, we investigated the therapeutic efficacy of L-DOS47 as a monotherapy and in combination with anti-PD1. Immunocompetent mice were inoculated orthotopically, and after tumor establishment (week 1, days 6–7), the mice were randomized into groups with equal tumor volume averages before initiating the therapies. The mice were treated twice a week, and tumor volumes were measured weekly via ultrasound (US) imaging to monitor tumor growth (Figure 4a).

Linear mixed-effects models were used to test for differences in tumor growth among the treatment arms. The tumor growth in all treatment arms was significantly greater than that for the combination treatment of anti-PD1 + L-DOS47 (Appendix A). Statistical analyses showed that the combination of anti-PD1 + L-DOS47 significantly reduced tumor growth when compared with the control (*p* = 0.01) and L-DOS47 monotherapy (*p* = 0.04) groups at week 2 (days 13–14), and it differed significantly from the anti-PD1 monotherapy at weeks 3 and 4 (days 20–21 and 27–28). Although the anti-PD1 monotherapy was effective in reducing tumor growth after 3 weeks when compared to the control group (*p* < 0.0001), the combination of anti-PD1 + L-DOS47 showed the greatest efficacy and had a synergistic effect when compared to the anti-PD1 monotherapy (*p* = 0.03 for week 2 and *p* = 0.01 for week 3) (Figure 4b–d; Table 1). The tumor growth plots for each experimental replicate are shown in Appendix A. In addition, the tumor weights measured at endpoint (week 4) were significantly lower in the anti-PD1 + L-DOS47 group when compared to the control (*p* = 0.01), L-DOS47 (*p* = 0.01), and anti-PD1 (*p* = 0.03) monotherapy groups (Figure 4e,f; Table 2).

## 4. Discussion

Although major advances have been made in other solid tumors, the utility of immunotherapy in PDAC has yet to be demonstrated. Various immunotherapies that have been investigated have proven largely unsuccessful, likely due to PDAC’s characteristically low tumor mutational burden and highly immunosuppressive tumor microenvironment [33]. 

Pembrolizumab, an anti-PD1 immune checkpoint inhibitor, has been approved by the FDA for a subset of patients with advanced PDAC whose tumors have been identified as mismatch repair deficient (dMMR) or microsatellite instability high (MSI-H), the latter of which accounts for only 0.8 to 2% of patients [33]. In the context of KEYNOTE-158, in a recent study of pembrolizumab in “all comers” with solid tumors, an 18.2% overall response rate (ORR), a 4.0-month median overall survival (OS), and a 13.4-month median duration of response were reported, with only three partial responses and one complete response in 22 PDAC patients with MSI-H or dMMR deficiency [34]. In a first dual approach with 65 patients with recurrent or metastatic PDAC, the PD-L1 antibody durvalumab in combination with the anti-cytotoxic T-lymphocyte antigen-4 (CTLA-4) antibody tremelimumab yielded only a 3.1% overall response rate compared to 0% for durvalumab monotherapy [35]. 

Acidosis is one of the major drivers and supporters of immune impairment during neoplastic formation, and it is imperative to target this condition for a successful treatment outcome [36]. In this study, we show in a murine orthotopic PDAC model that L-DOS47, an immunoconjugate that binds specifically to CEACAM6-expressing tumor cells, counters acidosis by raising pHe locally through the ureolytic activity of its urease enzyme moiety. L-DOS47 also acts synergistically with anti-PD1 therapy to slow tumor growth in this model. 

Previously, the specificity and cytotoxicity of L-DOS47 were confirmed in different CEACAM6-expressing cancer cell lines (BxPC-3 pancreatic, A549 lung, MCF7 breast, and CEACAM6-transfected H23 lung), wherein the response to L-DOS47 was positively correlated with the levels of CEACAM6 expression [19]. Indeed, L-DOS47 was most effective in reducing the in vitro viability of BxPC3 cells, which had the greatest levels of CEACAM6 expression [19]. Furthermore, tumor growth was significantly inhibited by L-DOS47 in a xenograft tumor model using BxPC3 [19]. This xenograft model utilized immunocompromised nude mice, and so any impact of L-DOS47 on the immune system could not be assessed [19]. The model we developed in this study using hCEACAM6-expressing KPC961 cells overcame this limitation and enabled the observation of the profound enhancement in anti-PD1 efficacy when combined with L-DOS47. 

Cellular metabolism was not altered during model establishment, and no spontaneous tumor rejection was observed in the control arms of our in vivo experiments, showing that no undue immunogenicity was caused by the expression of the human CEACAM6 protein. Moreover, once tumors were well established, they manifested the hallmarks of low pH as measured by CEST-MRI, which confirmed the utility of the model for testing anti-acidosis therapies, particularly L-DOS47. 

L-DOS47 was designed as a novel variation of antibody-directed enzyme prodrug therapy (ADEPT), in which the antibody component targets the molecule specific to its antigen—in this case, CEACAM6—on the tumor cell surface; however, unlike conventional ADEPT, in which prodrugs are administered systemically for the enzyme component to act upon, L-DOS47 uses the metabolite urea as a substrate, which is constitutively present in tumor tissues [37]. The urease enzyme component of L-DOS47 converts endogenous extracellular urea into ammonia and CO_2_, resulting in the formation of bicarbonate and hydroxyl ions, which alkalize the tumor microenvironment [19]. The pharmacodynamic studies using CEST-MRI described herein revealed a significant alkalinization of tumor pHe after a single dose of L-DOS47 (90 µg/kg) at all but one time point (from 4 to 96 h post administration, with the exception of 18 h), specifically in acidic tumors (basal pHe ≤ 6.60). 

In the in vivo therapeutic efficacy studies, optimal effects were observed with twice-weekly administration of L-DOS47 with anti-PD1, as the combination group exhibited significantly lower tumor volumes and weights compared to either monotherapy group. A significant difference was found at week 2 when comparing the combination therapy group to the control and L-DOS47 monotherapy groups. Strikingly, L-DOS47 together with anti-PD-1 was significantly more effective in controlling tumor growth than anti-PD1 alone. In the experiments shown here, anti-PD1 was administered 4 h before L-DOS47. Since anti-PD1 is known to have a long circulating half-life, and dosing was administered twice weekly, anti-PD1 would have been present in the mice as of the first treatment and, as such, if a difference was to be found, it would only be applicable to the first dose. Al-though the experimental design precluded the measurement of tumor pHe in the efficacy studies, our CEST-MRI results suggest a potential means of identifying L-DOS47 responders as those who present with acidic tumors with pHe ≤ 6.60.

Surprisingly, anti-PD1 monotherapy also provided some efficacy in our in vivo model, which is not commonly observed in patients with pancreatic cancer [38]. However, some patients with MSI-H tumors eventually respond to anti-PD1 with a similar survival rate to that achieved with the standard treatment for pancreatic cancer [38]. In line with this, it has been postulated that the immunologic status quo of the pancreatic tumor microenvironment can drastically differ across patients, where a pro-inflammatory state seems to adjuvate anti-PD1 efficacy [39,40]. While additional studies are warranted to evaluate the effects of L-DOS47 administration on immune cell subsets in the tumor microenvironment, it is nonetheless clear that the combination of L-DOS47 with anti-PD1 remained the most effective treatment overall in this study. 

Other preclinical studies have demonstrated that buffer systems, such as sodium bicarbonate, can alkalinize tumor pH and reverse the consequences of acidosis [41]. Oral administration of sodium bicarbonate prevented tumor development [42] and reduced invasion and metastasis in various tumor models, although it had no effect on the primary tumor growth [9,43]. We have also previously shown that buffering the tumor pH using sodium bicarbonate can improve the antitumor response to immune checkpoint therapies, as well as the adoptive transfer of T lymphocytes in B16 melanoma and Panc02 pancreatic tumor models. Combination therapy with bicarbonate and anti-PD1 or anti-CTLA-4 impaired tumor growth and led, in some cases, to tumor regression [15].

Unfortunately, these preclinical findings have not been supported by positive clinical trial results. The first three clinical trials using oral sodium bicarbonate (NCT01350583, NCT01198821, and NCT01846429) failed mainly due to the poor patient compliance associated with its unpleasant taste and/or gastrointestinal side effects [44]. More recently, patients with advanced pancreatic cancer (UMIN000035659) [45] and small-cell lung cancer (UMIN000043056) [46] did show improved outcomes after receiving an alkalization treatment, which included an alkaline diet and/or oral sodium bicarbonate (3.0–5.0 g/day). However, these were retrospective analyses of non-randomized single-center studies that included only a small number of patients [47].

Conversely, L-DOS47 has been proven safe and well tolerated in phase I/IIa clinical trials in heavily pretreated NSCLC patients, both as a monotherapy and in combination with pemetrexed and carboplatin [20,48]. Encouraging progression-free survival and clinical benefit were observed, particularly in patients who were also receiving pemetrexed/carboplatin (41.7% in objective response rate and 75% in clinical benefit). Additionally, a number of patients continued the L-DOS47 monotherapy well past the four protocol-mandated cycles. The maximum tolerated dose was not reached in either of the completed studies at doses of up to 13.55 µg/kg in the monotherapy and 9.0 µg/kg in the combination therapy study, which were both above the human equivalent dose of L-DOS47 used in the current study (7.3 μg/kg).

With the high expression of CEACAM6, L-DOS47 could be an ideal solution for treating cancers, such as lung, gastric, colorectal, and pancreatic cancers, in combination with chemo-, immuno-, and radiotherapies, as well as other therapeutic modalities, including cell and oncolytic viral therapies where acidosis is a limiting factor for efficacy [21,22]. Indeed, a phase I/II trial is currently ongoing to evaluate L-DOS47 in combination with doxorubicin in advanced pancreatic cancer patients (NCT04203641).

In this preclinical study, the reduction in tumor acidosis with L-DOS47 strengthened the anti-tumor response to anti-PD1 treatment by providing significantly greater tumor control. Future studies could test the efficacy of combining L-DOS47 and anti-PD1 or other immunotherapies in additional CEACAM6-expressing preclinical cancer models and clinical trials. In conclusion, L-DOS47 offers significant potential for broad applicability in combination with a growing number of innovative cancer treatments in the future.

## 5. Conclusions

Acidosis contributes to cancer progression by inhibiting anti-tumor immunity. We demonstrated that L-DOS47 can neutralize acidic tumor pH, which strengthens the anti-PD1 response in a PDAC preclinical model. These results, combined with the positive results of previous clinical trials, demonstrate that L-DOS47 is a clinically translatable agent to be considered as a novel therapy for PDAC patients. Given the current paucity of effective treatments and the poor response rates among PDAC patients approved for anti-PD1 therapy, our results suggest that adding L-DOS47 to this regimen could increase tumor pH and improve the outcomes for these patients. In addition, the study provides a hint on how crucial is to have a biomarker (pH in this case) that can lead to clinical translation based on the response to the drug. Furthermore, if efforts to render PDAC tumors more sensitive to immunotherapies are successful, this combination therapy may prove beneficial for all PDAC patients.

## Figures and Tables

**Figure 1 biomedicines-12-00461-f001:**
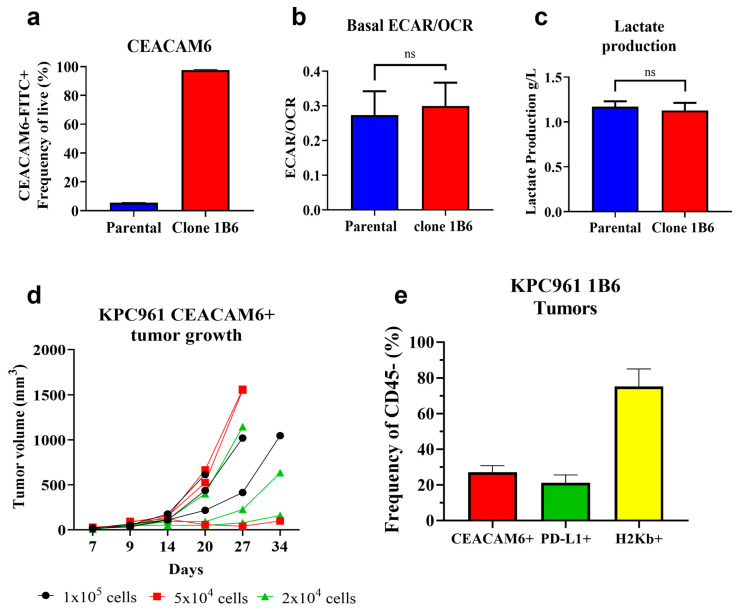
Generation of an orthotopic pancreatic adenocarcinoma murine tumor model expressing human CEACAM6. (**a**) Percentage of CEACAM6-expressing cells analyzed via flow cytometry in KPC961 parental and KPC961 clone 1B6 cells. (**b**) Ratio of extracellular acidification (ECAR) and oxygen consumption rate (OCR) profiles in KPC961 parental and clone 1B6 cells. (**c**) Lactate production in KPC961 parental and clone 1B6 cells. (**d**) Tumor growth as a function of inoculum size. KPC961-CEACAM6 cells were inoculated into the pancreas of B6.129 mice (n = 3 for each inoculum size) at 100,000, 50,000, and 20,000 cells, and tumor volumes were measured via ultrasound. Ascites fluid was observed in two mice inoculated with 100,000 cells on days 24 and 29, two mice inoculated with 50,000 cells on day 29, and one mouse inoculated with 20,000 cells on day 27. (**e**) Expression of CEACAM6, PD-L1, and H2Kb in KPC961-1B6 cells of inoculated orthotopic tumors obtained via flow cytometry.

**Figure 2 biomedicines-12-00461-f002:**
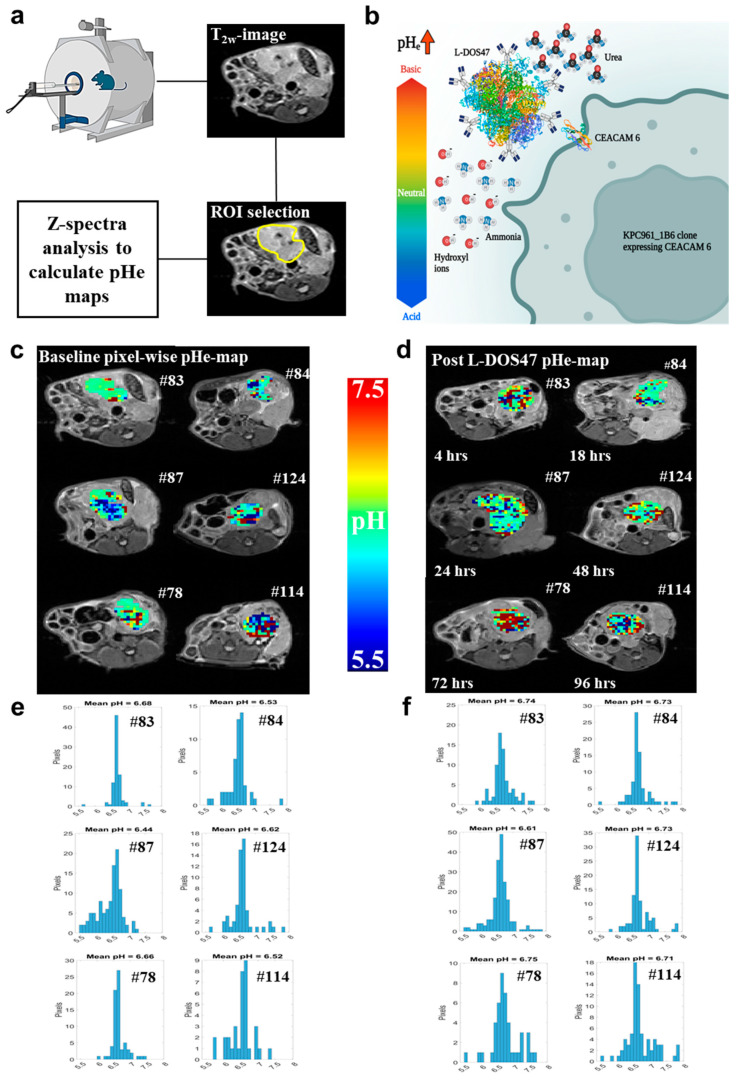
Pharmacodynamics of L-DOS47 in vivo as measured via MRI-CEST pH imaging. (**a**) Summarized scheme for image acquisitions and post-processing analyses for extracellular tumor pH (pHe) maps. A mouse was placed into the MRI scanner to start image collection, and a T_2w_ image was used as an anatomical (reference) image through which the region of interest (ROI) was drawn to selectively calculate pHe only in the tumor region of the CEST images. (**b**) L-DOS47 mechanism of action. Both urease and CEACAM6 protein structures were taken from the Protein Data Bank. The MRI mouse picture in (**a**) and the entire picture (**b**) were generated with BioRender.com. (**c**,**d**) Representative pHe maps from one mouse for each group (baseline and post L-DOS47). The pHe map was digitally superimposed onto the anatomical image. (**e**,**f**) Corresponding pHe map histograms showing the distribution of pHe values as a function of number of pixels across the ROI.

**Figure 3 biomedicines-12-00461-f003:**
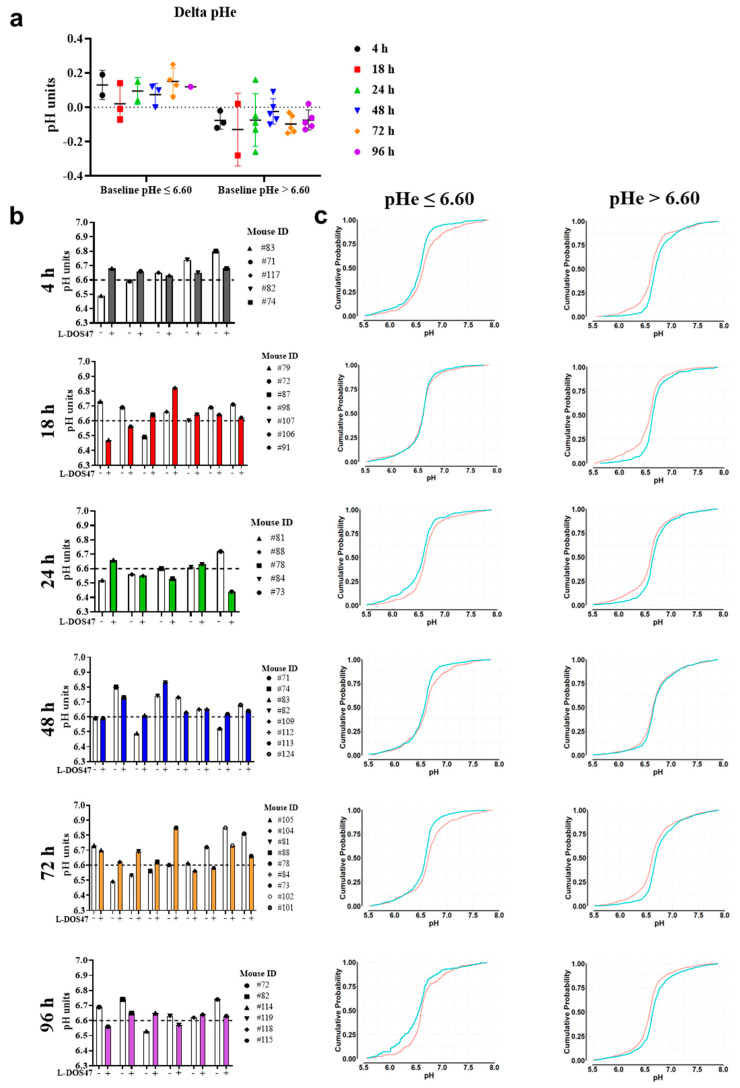
Single L-DOS47 administration effect on tumor pHe in vivo. (**a**) Dot plot of delta pHe values representing the pHe changes in each mouse in the different groups after being clustered into “responders” vs. “non-responders” (baseline pHe ≤ 6.60 vs. baseline pHe > 6.60, respectively). (**b**) Column bar plots reporting each mouse in each group with the baseline and post-treatment mean pHe (white-colored bars for the baseline values, and colored bars for the post-treatment values). (**c**) For every time point, the ECDF graphs are shown, and mice are clustered again into “responders” vs. “non-responders” to observe the pH shifts from the baseline (baseline shown as blue curve, and post-treatment shown as red curve). Refer to Appendix A for statistical significance.

**Figure 4 biomedicines-12-00461-f004:**
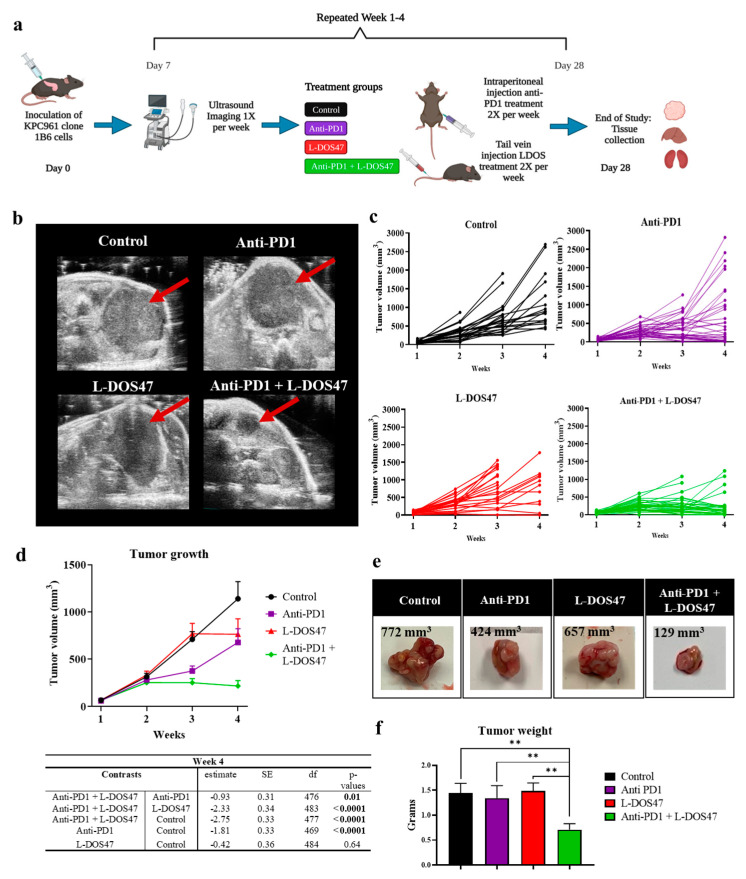
Therapeutic efficacy in KPC961-1B6 orthotopic tumors. (**a**) Experimental design. (**b**) Representative ultrasound images of KPC961-1B6 pancreatic tumors. (**c**) Individual tumor growth in each therapy group for 3 replicates (n = 28 in the control group; n = 35 in the anti-PD1 group; n = 23 in the L-DOS47 group; and n = 35 in the anti-PD1 + L-DOS47 group). (**d**) Mean tumor volume ± SEM for each therapy group with the corresponding table showing the differences at endpoint (week 4). (**e**) Representative ex vivo images of KPC961-1B6 tumors for each therapy group with the corresponding tumor volumes as measured by US (rounded to integral digit). (**f**) Mean tumor weight at endpoint. Statistical significance is reported based on the *p*-values obtained from the Tukey’s test ** *p* ≤ 0.01.

**Table 1 biomedicines-12-00461-t001:** Post hoc comparison of estimated marginal means based on the linear mixed-effects model for tumor growth in each therapy group (combined replicates 1, 2, and 3).

**Day 13**					
**Contrasts**		Estimate	SE	df	*p*-Values
Anti-PD1 + L-DOS47	Anti-PD1	−0.26	0.22	353	0.65
Anti-PD1 + L-DOS47	L-DOS47	−0.66	0.25	405	**0.04**
Anti-PD1 + L-DOS47	Control	−0.74	0.24	363	**0.01**
Anti-PD1	Control	−0.48	0.25	334	0.21
L-DOS47	Control	−0.08	0.26	432	0.98
**Day 21**					
**Contrasts**		estimate	SE	df	*p*-values
Anti-PD1 + L-DOS47	Anti-PD1	−0.60	0.22	340	**0.03**
Anti-PD1 + L-DOS47	L-DOS47	−1.49	0.24	389	**<0.0001**
Anti-PD1 + L-DOS47	Control	−1.75	0.24	347	**<0.0001**
Anti-PD1	Control	−1.15	0.24	316	**<0.0001**
L-DOS47	Control	−0.25	0.25	418	0.75
**Day 27**					
**Contrasts**		estimate	SE	df	*p*-values
Anti-PD1 + L-DOS47	Anti-PD1	−0.93	0.31	476	**0.01**
Anti-PD1 + L-DOS47	L-DOS47	−2.33	0.34	483	**<0.0001**
Anti-PD1 + L-DOS47	Control	−2.75	0.33	477	**<0.0001**
Anti-PD1	Control	−1.81	0.33	469	**<0.0001**
L-DOS47	Control	−0.42	0.36	484	0.64

Degrees-of-freedom method: Kenward–Roger. Results are given on the log2 (not the response) scale. *p*-Value adjustment: Tukey’s method for comparing a family of 4 estimates. The bold numbers are the significant *p* values.

**Table 2 biomedicines-12-00461-t002:** Tukey’s Honest Significant Difference post hoc tests and pairwise comparisons of tumor weight marginal means.

Contrasts		Estimate	SE	df	*p*-Values
Control	L-DOS47	−0.04	0.28	−0.15	0.99
Control	Anti-PD1	0.10	0.27	0.37	0.98
Control	Anti-PD1 + L-DOS47	0.73	0.24	3.05	**0.01**
L-DOS47	Anti-PD1 + L-DOS47	0.77	0.24	3.17	**0.01**
Anti-PD1	Anti-PD1 + L-DOS47	0.63	0.23	2.73	**0.03**

*p*-Value adjustment: Tukey’s method for comparing a family of 4 estimates. The bold numbers are the significant *p* values.

## Data Availability

All materials (data and images) reported in this article are available within the paper and its Appendix A.

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
