# Peer review of "L-DOS47 Elevates Pancreatic Cancer Tumor pH and Enhances Response to Immunotherapy"

_biomedicines, 2024, doi:10.3390/biomedicines12020461_

Round 1
Reviewer 1 Report
Comments and Suggestions for Authors
I think that this paper is quite valuable. However, necessary corrections should be made before it is accepted for publication.
1. Why is there no mention of pancreatic cancer in the introduction? Such an introduction is necessary, so please fill in this missing part.
2. Why have this cell line been chosen? I think at least two should have been used (please do not count the control line here). Please explain exactly why this choice was made.
3. Please explain your conclusions in more detail. What is the outcome of your experiments or is there any chance that this molecule could be used for medical purposes in the future?
Author Response
Thank you for giving us the opportunity to submit a revised draft of the manuscript. We appreciate the time and effort that you dedicated to providing feedback on our manuscript and are grateful for the insightful comments on and valuable improvements to our paper. We have incorporated most of the suggestions. Those changes are highlighted within the manuscript. Please see a, in the attached file, for a point-by-point response to comments and concerns.

Reviewer 2 Report
Comments and Suggestions for Authors
In the manuscript titled "L-DOS47 Elevates Pancreatic Cancer Tumor pH and Enhances Response to Immunotherapy," the authors present a compelling investigation into the integration of immunotherapy with complementary treatment modalities, notably addressing acidosis, to enhance therapeutic efficacy. This work reflects a well-documented and meticulously executed study, showcasing the authors' adeptness in selecting an appropriate murine model and systematically comparing their treatment approach against existing modalities while highlighting the novelty of their discoveries. While the manuscript is well-detailed, there are some suggestions made that are aimed to enhance enthusiasm for publication.
-
How L-DOS47 targets tumors specifically while avoiding normal epithelial linings, considering CEACAM6's presence in various organ linings such as GI, breast etc. under normal conditions.
-
Considering the potential impact of excess NH4+ ion production, exploring the impact on liver enzyme levels would be a valuable addition. Were any murine liver samples collected or molecular studies conducted to investigate this aspect further? If not, it may be worth exploring.
-
Studying the long-term effects of excessive ammonium ion production is crucial. Due to excessive production of ammonium ions it would be a good idea to study the effects long term.
Minor changs:
-
Please ensure consistency in figure callouts between the main text and figure legends to enhance readability. Choose one style (capital or small letters) and make sure it is consistent in texts and legends.
-
Incorporating higher resolution images or graphs for clarity, particularly regarding images 2e, 2f, 4a, and 4b, would be reader compliant.
-
Sentence is incomplete L 380.
Author Response

(The authors gave the same response as above.)

Reviewer 3 Report
Comments and Suggestions for Authors
The manuscript deals with the preclinical investigation of a novel pH-modulating drug alone and in combination with pembrolizumab in an immunocompetent mice model to evaluate the combination strategy and the role of pH in tumor progression in the pancreatic tumor. But Some points need to be clarified.
General comment': Pembrolizumab is very reactive in humans and mice with ADR immuno-based including cytokine release syndrome targeting different tissues, but these were not observed in this animal model. The animal model needs to be better detailed.
Methods Line 243 the phrase is not clear: Mice that developed an additional tumor in the abdominal wall were excluded from the study,
Are these tumors or immunoreactions?
It is necessary to add a separate cell toxicity paragraph related to the effects of the human cancer cell in the control mice not treated with any drugs to evaluate toxicity related to autoimmune reactions using an appropriate sample size that can be different based on the incidence of the ADR that you are looking for.
Methods: In this work, the sample size was not reported based on the primary endpoint and also based on autoimmune reactions
Line 441: Cellular metabolism was not altered during model establishment, and no spontaneous tumor rejection was observed in control arms of our in vivo experiments, showing that no immunogenicity was caused by expression of the human CEACAM6 protein,
This sentence needs to be supported by data. How many mice have you injected ???
Author Response

(The authors gave the same response as above.)

Round 2
Reviewer 1 Report
Comments and Suggestions for Authors
The authors reply to the comments
Reviewer 3 Report
Comments and Suggestions for Authors
The manuscript has been improved